# Comparison of the Aroma-Active Compounds and Sensory Characteristics of Different Grades of Light-Flavor Baijiu

**DOI:** 10.3390/foods12061238

**Published:** 2023-03-14

**Authors:** Huanhuan Li, Xin Zhang, Xiaojuan Gao, Xiaoxuan Shi, Shuang Chen, Yan Xu, Ke Tang

**Affiliations:** 1Lab of Brewing Microbiology and Applied Enzymology, School of Biotechnology, Jiangnan University, 1800 Lihu Avenue, Wuxi 214122, China; 2Shanxi Xinghuacun Fenjiu Distillery Co., Ltd., Fenyang 032205, China; 3Shanxi Provincial Key Laboratory for Chinese Lujiu Plant Extraction and Health, Fenyang 032205, China

**Keywords:** light-flavor Baijiu (LFB), grades, GC-O-MS, odor activity values (OAVs), sensory evalution

## Abstract

This study comprehensively characterized and compared the aroma differences between four different grades of Fenjiu (FJ, the most representative light-flavor Baijiu). Aroma-active compounds were analyzed by liquid-liquid extraction (LLE) coupled with gas chromatography-olfactometry-mass spectrometry (GC-O-MS). A total of 88 aroma-active compounds were identified, and 70 of them were quantified. The results showed that a majority of aroma compounds in high-grade FJ had higher aroma intensities and concentrations. Among these compounds, there were 28 compounds with odor activity values (OAVs) greater than one in all four wines, which indicated that they might contribute to the characteristic aroma of FJ. Temporal dominance of sensation (TDS) and quantitative descriptive analysis (QDA) were used to characterize the sensory differences. The results suggested that high-grade FJ had a rich, pleasant and lasting retronasal aroma perception and exhibited pleasant orthonasal aroma of floral, fruity, sweet and grassy. Partial least squares regression (PLSR) analysis effectively distinguished four kinds of FJ and revealed associations between the orthonasal aroma attributes and the aroma compounds with OAVs >1. There were 15 compounds with variable importance in projection (VIP) values >1, and they were considered potential aroma markers for quality prediction.

## 1. Introduction

Baijiu is a traditional fermented and distilled Chinese alcoholic beverage. It has a broad market and a large number of consumers and constitutes the largest segment of the Chinese alcoholic beverage industry [1]. In 2021, 7.156 billion liters of Baijiu were produced, with a sales revenue of USD 89.409 billion [2]. Quality determines the value of Baijiu, and the flavor is the core element of quality [3]. The unique manufacturing process produces a unique flavor of Baijiu. To produce Baijiu, raw materials need to go through complex solid-state fermentation under the influence of the microorganisms in Jiuqu, and then base liquor is produced following the Zeng distillation process [4]. Before it is sold in the market, aging and blending are required to transform the base liquors into commercial products, and the grade of these commercial products is determined on the basis of their flavor [5]. However, the flavor grading process of Baijiu is sometimes subjective and unstable since it mainly depends on the individual experience of professional tasters. In order to objectively evaluate the quality and increase the yield of high-quality Baijiu, it is necessary to characterize the flavor characteristics of different grades of Baijiu and identify their corresponding key compounds [6].

Aroma is one of the important factors affecting the quality and consumer acceptance of Baijiu [7]. On the basis of the differences in aroma characteristics, Baijiu is generally divided into twelve types [8]. Among them, light-flavor Baijiu (LFB) is well-known for its elegant and pure aroma, particularly in northern China [9]. Fenjiu (FJ), as the most typical representative liquor of LFB, is widely popular among consumers due to its pleasant fruity and floral aroma. The aroma of Baijiu largely depends on the aroma-active compounds dissolved in it. The characterization of aroma-active compounds in LFB has never stopped. Gao et al. [9] found that *β*-damascenone and ethyl acetate were the key odorants of LFB, and Niu et al. [10] considered that dimethyl trisulfide and 3-mercaptohexyl acetate were the key sulfur compounds. Wang et al. [11] found five unreported aroma-active compounds in LFB and explored the changes in the key odorants’ contents during the storage of LFB. Although many studies have been carried out to find the important aroma-active compounds in LFB, the characteristics and differences of these compounds in LFB with different qualities are still unclear.

To distinguish the different grades of LFB, characterization and comparison of aroma-active compounds in LFB is one of the critical steps. In previous studies, gas chromatography-mass spectrometry (GC-MS) combined with gas chromatography-olfactometry (GC-O) has been proven to be a valuable tool to characterize the aroma-active compounds in Baijiu [9,10,11,12,13]. Osme and aroma extract dilution analysis (AEDA) were the most commonly used method in GC-O analysis [14]. In addition to instrumental analysis, sensory analysis is also an important method to characterize aroma quality. The classic and most employed sensory technique in Baijiu is quantitative descriptive analysis (QDA), which can not only describe the sensory characteristics of products but also distinguish the sensory differences between different samples [15,16]. However, this method can only assess the perception of aroma as a “static” phenomenon. Since Baijiu is ultimately consumed in the mouth, the retronasal aroma is considered to play a key role in determining the final flavor perception of products, and the flavor perception during baijiu consumption is a dynamic and complex process. The dynamic sensory evaluation of alcoholic beverages is generally considered a difficult task, mainly due to the complex aroma and high alcohol content [17]. Temporal sensory methodologies such as time-intensity (TI) and temporal dominance of sensation (TDS) have been used to describe the temporal profiles of the pungency of Baijiu with different aging times [18]. Nevertheless, few studies have focused on the dynamic assessment of the retronasal aroma perception of Baijiu.

Accordingly, the aim of this study was: (1) to identify the major aroma-active compounds in different grades of FJ using liquid-liquid extraction (LLE) coupled with GC-MS and GC-O; (2) to quantity those aroma-active compounds by using multiple methods; (3) to evaluate the contribution of those aroma-active compounds by calculating their OAVs; (4) to compare the orthonasal and retronasal aroma differences between the four different grades of FJ by QDA and TDS; (5) to predict the potential aroma quality-markers by partial least squares regression (PLSR) analysis. The findings obtained in this study should be helpful in understanding the flavor characteristics of LFB and further improving the aroma quality of LFB during production.

## 2. Materials and Methods

### 2.1. Baijiu Samples

Four different grades of FJ (labeled as F00, F10, F20, and F30, with an alcohol content of 53%, *v*/*v*) were chosen as the samples, which were provided by Shanxi Xinghuacun Fenjiu Group Co., Ltd., Fenyang, China. They were manufactured from the same materials and production process, but the base liquors they used to blend were different. The aging time of their base liquors was 0, 10, 20, and 30 years respectively. The grade of samples increased with the aging time of their base liquor. These liquors were all stored at 4 °C before analysis.

### 2.2. Chemicals

The analytical standards used to identify the aroma compounds were of high-purity grade (GC grade, ≥98%). Ethyl acetate, ethyl propanoate, ethyl 2-methylpropanoate, 2-methylpropyl acetate, ethyl butanoate, ethyl 2-methylbutanoate, ethyl 3-methylbutanoate, 3-methylbutyl acetate, ethyl pentanoate, ethyl hexanoate, hexyl acetate, 3-methylbutyl butanoate, ethyl heptanoate, ethyl lactate, ethyl octanoate, 3-methylbutyl hexanoate, octyl acetate, ethyl nonanoate, diethyl malonate, ethyl decanoate, ethyl benzoate, diethyl butanedioate, ethyl phenylacetate, 2-phenylethyl acetate, ethyl dodecanoate, ethyl 3-phenylpropanoate, ethyl tetradecanoate, 2-butanol, 1-propanol, 2-methylpropanol, 1-butanol, 3-methylbutanol, 1-pentanol, 2-heptanol, 1-hexanol, 3-hexen-1-ol, (*Z*)-, 1-heptanol, 1-octen-3-ol, 1-octanol, 2,3-butanediol, 1-nonanol, 2-phenylethanol, acetic acid, propanoic acid, 2-methylpropanoic acid, butanoic acid, 2-methylbutanoic acid, 3-methylbutanoic acid, pentanoic acid, hexanoic acid, octanoic acid, decanoic acid, 2-methylpropanal, 3-methylbutanal, hexanal, nonanal, decanal, benzaldehyde, (*E*,*Z*)-2,6-nonadienal, phenylacetaldehyde, naphthalene, 2-pentylfuran, 2-furaldehyde diethyl acetal, furfural, 5-methyl furfural, 2-furanmethanol, 2(5H)-furanone, 2-heptanone, 3-hydroxy-2-butanone, 2-nonanone, γ-hexalactone, γ-nonalactone, guaiacol, 4-methylguaiacol, phenol, 4-ethylguaiacol, 4-ethylphenol, 2,6-dimethylpyrazine, trimethylpyrazine, tetramethylpyrazine, dimethyl disulfide, dimethyl trisulfide, 2-thiophenecarboxaldehyde, *β*-damascenone, geosmin, geranylacetone, *β*-ionone, 1,1-diethoxyethane were obtained commercially from Sigma-Aldrich Co., Ltd. (Shanghai, China), J&K Scientific Co., Ltd. (Shanghai, China), and Alfa Aesar (Tianjin, China). A mixture of C_6_–C_30_ hydrocarbons (Sigma-Aldrich, Shanghai, China) was employed to determine linear retention indices (RIs). Sodium chloride (NaCl) and anhydrous sodium sulfate (Na_2_SO_4_) were acquired from China National Pharmaceutical Group Corp (Shanghai, China). Ethanol (high-performance liquid chromatography (HPLC) grade) and dichloromethane (CH_2_Cl_2_, HPLC grade) were bought from Titan Scientific Co., Ltd. (Shanghai, China). Ultrapure water was obtained from a Milli-Q purification system (Millipore, Bedford, MA, USA). 2-Methyl-2-butanol (IS1), n-pentyl acetate (IS2), dl-menthol (IS3), octyl propanoate (IS4), trimethylacetic acid (IS5) (Sigma-Aldrich, Shanghai, China) were used as internal standards in quantitative analysis. 

The reagents used in QDA were high-purity grade (GC grade, ≥98%), while the references of the retronasal aroma were food grade (except for geosmin, ≥95%). They were all purchased from Sigma-Aldrich (Shanghai, China). Purified water (Wahaha Group Co., Ltd., Hangzhou, China) was used to clean the mouth between samples in sensory analysis.

### 2.3. Identification of Aroma-Active Compounds

#### 2.3.1. Aroma Extraction by LLE

According to a method from a previous study [19], four different grades of FJ (50 mL each) were diluted to 10% (*v*/*v*) with boiled ultrapure water. Then they were saturated with NaCl and extracted three times with 50 mL CH_2_Cl_2_. The combined extracts were further dried with anhydrous Na_2_SO_4_ overnight and then concentrated to a final volume of 500 μL under a gentle stream of nitrogen. The concentrated fractions were stored at −20 °C before the GC−O analysis.

#### 2.3.2. Identification of Aroma-Active Compounds by GC-O-MS

GC-O-MS analysis was performed on an Agilent 7890B GC equipped with an Agilent 5977B mass selective detector and a sniffing port (ODP 4, Gerstel, Württemberg, Germany). The samples were analyzed on both a DB-FFAP column (60 m × 0.25 mm × 0.25 μm, Agilent Technologies, Inc., Santa Clara, CA, USA) and an HP-5 column (30 m × 0.25 mm × 0.25 μm, Agilent Technologies, Inc., Santa Clara, CA, USA). The carrier gas was high-purity helium (>99.999%) at a flow rate of 1.8 mL/min. The concentrated extract (1 μL) was injected into the GC injector at 250 °C in splitless mode. The column temperature program was set as follows: the oven temperature was initially held at 45 °C for 2 min, then raised to 230 °C at the rate of 6 °C/min, and held for 15 min. The temperature of the ion source was 230 °C, and the ionization energy was 70 eV. The range of masses full-scan was used in the 35–350 amu, and a 5 min solvent delay was applied.

Four trained panelists (2 males and 2 females, 25 years old on average) from the Laboratory of Brewing Microbiology and Applied Enzymology at Jiangnan University were employed for GC-O analysis. During a GC run described above, a panelist placed his/her nose close to the sniffing pore and recorded the retention time, aroma descriptors and intensity value of the odor peak for each compound. A 6-point scale (0, 1, 2, 3, 4, 5) was used for intensity judgment: 0 = none, 1 = very weak, 2 = weak, 3 = moderate, 4 = strong, and 5 = very strong. Every sample was repeated twice. The Osme value was the average result of the 8 records. The compound could be confirmed only when the aroma was perceived 6 or more times by the panelists. Aroma-active compounds were identified by comparison of their mass spectra, odors, and RI with the corresponding pure standards. The RI values were calculated based on the linear retention times of n-alkanes (C_6_–C_30_) in both the DB-FFAP and HP-5 columns under the same chromatographic conditions.

### 2.4. Quantitation of Aroma-Active Compounds

Considering the complexity of aroma-active compounds in Baijiu, multiple methods were used to conduct the quantitative analysis. Gas Chromatography with Flame Ionization Detection (GC-FID) was used for the quantitation of some compounds with high concentrations. Headspace Solid-Phase Microextraction-Gas Chromatography-Mass Spectrometry (HS-SPME-GC-MS) was used to quantify a majority of aroma compounds. Due to the strong polarity of fatty acids, SPME fiber can not adsorb these compounds well. So Liquid−Liquid Microextraction (LLME) was used for the quantitation of these acids. The calibration curve with at least six concentration levels for each standard compound was built up with a 1:1 dilution series in aqueous ethanol (10%, *v*/*v*). The methods of extraction and detection were the same as those of the samples. All samples were analyzed in triplicate.

#### 2.4.1. GC-FID

1,1-Diethoxyethane, ethyl acetate, 1-propanol, 2-methylpropanol, 3-methylbutanol and ethyl lactate were quantified by an Agilent (Santa Clara, CA, USA) 7890A GC system, equipped with a flame ionization detector. Baijiu samples (1 μL) with internal standards (IS1: 2-methyl-2-butanol, 116.59 mg/L, IS2: n-pentyl acetate, 131.78 mg/L) added were directly inserted into the GC system in split mode (split ratio = 20:1). Hydrogen was used to operate FID at a flow rate of 30 mL/min. A CP-WAX column (50 m × 0.25 mm × 0.20 μm, Agilent Technologies, Inc., Santa Clara, CA, USA) was used for the separations. High-purity helium (>99.999%) was used as the carrier gas at a constant flow rate of 1 mL/min. The oven temperature was initially held at 35 °C for 5 min, then raised to 100 °C at the rate of 4 °C/min, and held for 2 min, then increased at 8 °C/min to 150 °C, and finally raised at 15 °C/min to 200 °C and held for 25 min. The injector and detector temperatures were set at 250 °C

#### 2.4.2. HS-SPME-GC-MS

The extraction and injection of the samples were conducted by an SPME-Arrow automatic headspace sampling system (CTC Analytics AG, Zwingen, Switzerland) with a 120 μm divinylbenzene/carbon wide range/polydimethylsiloxane (DVB/CAR WR/PDMS) fiber (CTC Analytics AG, Basel, Switzerland). Five mL of diluted Baijiu sample (10%, *v*/*v*) and 1.5 g of NaCl were added to a 20 mL glass headspace vial. IS3 (dl-menthol, 90 μg/L) and IS4 (octyl propanoate, 119.11 μg/L) were used as internal standards. The sample was equilibrated at 45 °C for 5 min and stirred at 250 rpm for 45 min at the same temperature. After extraction, the fiber was inserted into the GC injector port (250 °C) for a 5 min desorption in spitless mode. The column temperature program was the same as that used for the GC-O-MS analysis previously described. 

#### 2.4.3. LLME

Diluted Baijiu samples (18 mL, 10%, *v*/*v*) with 6 μL of IS5 solution (trimethylacetic acid, 3406.43 μg/L) added were saturated with NaCl (6 g) and then shaken for 5 min with 1.5 mL of redistilled diethyl ether added. After standing for a period of time, the upper extracts were sucked out and then concentrated to 250 μL under a gentle stream of nitrogen. Finally, 1 μL of concentrate was injected into the injection port of the GC (250 °C) for analysis.

### 2.5. Sensory Evaluation

#### 2.5.1. Panel Selection and Training

Voluntary registration was used to recruit panelists among the students of Jiangnan University according to the guidelines of the ISO 8586:2012 standard [20]. Personal information (name, gender, age, contact information, health status, allergic history and spare time) was collected by filling out a questionnaire. The panelists who were in good health, non-smoking and had no allergic history of alcohol or other aroma compounds were selected. Then they were asked to participate in the discrimination, ranking and recognition tests. Twenty-five candidates who had achieved at least 70% acuity were selected finally. They were paid for their involvement.

The general training (24 sessions, 1 h/session, twice a week) mainly focused on the basic sensory perception training of Baijiu, including the aroma description and identification, ranking, and triangle tests. Panelcheck was used to monitor the performances of the panelists and ensure the effectiveness of the training. Finally, 16 panelists (6 female and 10 male, aged between 19 and 25 years old) who could clearly describe and perceive both the orthonasal and retronasal aromas of different liquors were selected to constitute the final sensory panel. 

#### 2.5.2. Temporal Dominance of Sensation (TDS)

To explore the changes and differences in the retronasal aroma perception of the samples during consumption, 16 panelists received a training period of 3 one h sessions to learn the principle of TDS, the definition of dominant retronasal aroma attributes, and the use of Sensomaker Software. Taste attributes and pungency were not considered during the evaluation process. After discussion, 8 retronasal aroma attributes of alcoholic, grain, acidic, fruity, floral, roasted, grassy and earthy were finally selected. The definitions and references of these 8 attributes were determined by consensus. The details are provided in Appendix A. 

The evaluation procedure of TDS is shown in Appendix A. To avoid any order effect, the order of the list of attributes was different for each panelist but the same for a given judge during the entire evaluation [21]. Panelists were required to put a 5 mL baijiu sample into their mouth, then click on the “start” button in the SensoMaker software. The delay time was set to 10 s. During this time, the panelists performed a soft rinsing and were not allowed to swallow or open the velum-tongue border. The liquor was spatted out when the delay time expired. Then, the panelists started to select the most dominant attribute. When they felt the perception had changed, the next most dominant attribute should be chosen. During the evaluation, the panelists were free to select an attribute several times or, conversely, to never choose an attribute as dominant. Finally, when sensations were no longer recognizable, panelists had to click on the “stop” button to end the recording. The data acquisition would automatically stop after 90 s. Evaluations took place in isolated sensory booths at a standardized temperature (20 °C ± 1 °C). All the samples were randomly coded and presented according to Williams Latin Square. Each sample was evaluated in triplicate. For further recovery, panelists were requested to break at least 10 min between samples. Rinsing their mouths with tap water and eating unsalted crackers to eliminate any traces of aroma were allowed.

#### 2.5.3. Quantitative Descriptive Analysis (QDA)

After the TDS assessment, 16 panelists continued to participate in the QDA training, including 12 sessions (1 h each). The first 3 sessions were used to determine the orthonasal aroma attributes and confirm their corresponding references. Seven descriptors with high perception frequency that can accurately describe the characteristics and differences of four samples were selected. The definitions and references are shown in Appendix A. During the next 9 sessions, an 8-point unstructured line scale (0–7) was used to evaluate the orthonasal aroma intensity in those samples. A supplementary scoring standard (Appendix A) was developed to help the panelists score. Finally, 10 panelists with good consistency, stability, and repeatability participated in the final QDA experiment. 

15 mL of Baijiu sample was served in standard Baijiu tasting glasses, covered and coded with random three-digit codes. Four samples were provided to the panelists at the same time. For a specific attribute, the panelists sorted these four samples first, then scored the intensity in turn. After evaluating 1 attribute, the next attribute could be assessed. The panelists were required to rest for at least 5 min between different attributes.

### 2.6. Statistical Analysis

Analysis of variance (ANOVA) was performed with SPSS 25.0 (SPSS, Inc., Chicago, IL, USA). The XLSTAT 2016 (Addinsoft, Paris, France) was used for principal component analysis (PCA) and partial least squares regression (PLSR) analysis. Panelcheck were used to monitor the performances of the panelists. The QDA data collected manually were analyzed by Microsoft Excel 2019. The TDS data were collected using SensoMaker software. GraphPad Prism 7.0 (GraphPad Software, San Diego, CA, USA) was used to draw TDS curves. The vertical drop lines of osme values and the aroma profile were drawn by OriginPro 2023 (OriginLab, Northampton, MA, USA).

## 3. Results

### 3.1. Identification of Aroma-Active Compounds

As shown in Table 1, a total of 88 aroma-active compounds were identified by LLE combined with GC-O-MS analysis in four different grades of FJ. A few aroma compounds could not be detected by mass spectrometry due to their low concentrations. But the panelists could clearly perceive the aroma. Then these compounds were inferred by referring to literature, comparing aroma and RIs at first, and finally identified by the authentic standards. In this study, it was first found that 2-heptanone, 2,6-dimethylpyrazine, 3-hexen-1-ol, (*Z*)-, octyl acetate, 5-methyl furfural and 2(5H)-furanone were the aroma-active compounds in LFB. However, their osme values were very low, indicating that they may make fewer contributions to the overall aroma of LFB. By comparison, F30 had the largest number of aroma compounds that could be perceived by the panelists, while F00 had the least. Figure 1 compared the osme values of these aroma compounds, which varied in different samples. The result showed that a majority of aroma compounds presented the highest intensity in F30 while having the lowest in F00. It could be found that high-grade FJ contained more aroma compounds and had higher aroma intensities, which is consistent with the study in different grades of sesame-flavor Baijiu [16].

There were only a few compounds in the low-grade samples (F00, F10) with osme values greater than 4 (none in F00 and 2 in F10). Nevertheless, in F30, 8 aroma compounds were considered to make the most important contributions to the overall aroma as their higher aroma intensity of at least four, which were 3-methylbutyl acetate (fruity), 3-methylbutanol (malty), ethyl hexanoate (fruity), ethyl octanoate (fruity), ethyl phenylacetate (rosy), 2-phenylethyl acetate (floral), *β*-damascenone (honey), and ethyl 3-phenylpropanoate (floral). The aroma intensities of ethyl acetate (pineapple), 2-methylpropanol (malty), 3-methylbutyl acetate (fruity), 3-methylbutanol (malty), ethyl hexanoate (fruity), dimethyl trisulfide (cabbage), tetramethylpyrazine (nutty), butanoic acid (sweaty), ethyl phenylacetate (rosy), 2-phenylethyl acetate (floral), *β*-damascenone (honey) and 4-ethylphenol (animal) were all beyond 3 in those four samples. Among these compounds, ethyl acetate, 3-methylbutyl acetate, dimethyl trisulfide, butanoic acid and *β*-damascenone have been proven to have a significant influence on the final aroma characteristics of LFB [9,10,11,13].

### 3.2. Quantitation of Aroma-Active Compounds 

Given the extremely low contents or low contributions and combined with the reports of literature, finally, 70 aroma compounds (26 esters, 12 alcohols, 10 acids, seven aldehydes, two ketones, five phenols, three terpenes, and five others) were quantified by constructing calibration curves of each one. The quantitative methodological parameters are given in Appendix A. The concentrations of aroma compounds varied in four different grades of FJ (Table 2). The contents of most aroma compounds in high-grade FJ were higher than those in low-grade FJ. Wang et al. [13] found that the content of esters, alcohols, and acids increased with the increase in the quality grades of Caoyuanwang (CYW, a kind of LFB), and Qin et al. [16] also suggested that the premium-grade sesame-flavor Baijiu had the highest aroma concentrations. The findings declared that high-grade Baijiu might have higher aroma concentrations. These may be due to the different aging years and blending proportions of their base liquors. The results of the Duncan test showed that ethyl octanoate, 3-methylbutyl hexanoate, ethyl decanoate, ethyl benzoate, ethyl dodecanoate, ethyl 3-phenylpropanoate, 2-heptanol, 1-octen-3-ol and 4-methylguaiacol had significant differences among four samples, which were likely to play a key role in differentiating the differences between different grades of FJ.

#### 3.2.1. Esters

Among these quantified odorants, the esters, which mainly contribute to the aroma of fruity, floral, and sweet Baijiu, had the most species and highest concentrations [22]. Esterification with alcohol and acid as substrates is the main pathway to produce most esters, which includes biochemical esterification and spontaneous chemical esterification [23]. The higher the sample grade, the higher the concentration of most of the esters. Eighteen esters had the highest concentrations in F30, followed by F20, F10, and F00. Ethyl acetate and ethyl lactate were the two esters with the highest concentrations in FJ, which played an important role in the quality of Baijiu [24,25,26]. However, the quantitation results showed that the concentrations of these two substances were relatively higher in the low-grade sample F00, indicating that the concentration is not the only factor determining the quality. For example, Wang et al. [26] have found that lactic acid could give additive or synergistic odor effects for the two esters, thus enhancing the fruity aroma. Ethyl 2-methylbutanoate and ethyl 3-methylbutanoate have been proven to be potentially important to the overall aroma profile of Qingke Jiu [27]. Ethyl phenylacetate, 2-phenylethyl acetate and ethyl 3-phenylpropanoate have positive contributions to the floral sensory profiles of Baijiu. Their contents in high-grade FJ were higher than those in low-grade FJ. Niu et al. [22] found that ethyl phenylacetate had a mask effect of fruity note while phenylethyl acetate added at peri-threshold (3200 ppb) would significantly enhance the sweet note.

#### 3.2.2. Alcohols

The alcohol could make the aroma and taste of the liquor harmonious and increase the sweetness and aftertaste of the liquor [28]. They are generally synthesized by yeast and other microorganisms via the amino acid catabolic pathway and the sugar metabolism synthetic pathway [29]. 3-Methylbutanol, 1-propanol and 2-methylpropanol were the top three alcohols with the highest concentrations in FJ, which mainly contributed to the alcoholic and malty aroma. They all had relatively higher concentrations in F20. 3-Methylbutanol has been recognized as one of the important aroma compounds in fresh Xiaoqu Baijiu [12]. 1-Propanol with a high concentration could significantly mask the aroma of 3-methylbutanoic acid [30]. The remaining nine alcohols all had the highest concentrations in F30, except for 1-Octanol (higher in F10). 2-Phenylethanol has a pleasant rose-like odor, which also has a high concentration in Chinese rice wine [31]. 1-Octen-3-ol is one of the main volatiles of light-flavor Daqu [32] and an important aroma-active compound to Qingke liquors [33]. 

#### 3.2.3. Acids

The acids are produced during the metabolism of ethanol and have a huge impact on the aroma, taste and function of Baijiu [34]. It has been studied that acids could contribute to pungency perception [35]. Acetic acid had the highest concentrations among all the acids, followed by propanoic acid, hexanoic acid and butanoic acid. Acetic acid is the main volatile organic acid in Baijiu, which has a significant contribution to the overall aroma of LFB [9]. The higher the sample grade, the higher the content of acetic acid. This is mainly due to the different storage times of the base liquors they used to blend. The higher the grade, the longer the aging time of base liquor. A previous study has confirmed that the content of acetic acid would increase significantly during the aging process of FJ [36]. 2-Methylbutanoic acid, hexanoic acid and octanoic acid had relatively higher concentrations in F30, while the concentrations of propanoic acid, 2-methylpropanoic acid, butanoic acid and pentanoic acid were higher in F20. 3-Methylbutanoic acid and decanoic acid had higher concentrations in F00. 3-Methylbutanoic acid mainly contributes to the aroma of sweaty, stinky, and cheesy Baijiu, which is always thought of as a kind of off-odor [30].

#### 3.2.4. Aldehydes and Ketones

Various sources could produce aldehydes and ketones, like alcohol oxidation, ketone acid decarboxylation, amino acid deamination, and decarboxylation metabolic pathways [6]. Seven aldehydes and two ketones were quantified. 3-Methylbutanal had the highest concentration, which was relatively higher in F10. It has been proven to be the key odorant in Qingke liquors [33] and has an important contribution to the aroma of Niulanshan Baijiu [11]. 2-Methylpropanal, nonanal, decanal, benzaldehyde and phenylacetaldehyde had higher concentrations in F30, while hexanal was higher in F20. The concentrations of 2-heptanone and 2-nonanone were higher in F00. 

#### 3.2.5. Phenols, Terpenes and Others

Among phenols, phenol had the highest concentrations, which was significantly higher in F10. Guaiacol, 4-methylguaiacol and 4-ethylguaiacol mainly contributed to the clove and smoky aroma. Guaiacol had a higher concentration in F10, while 4-methylguaiacol was much higher in F30. 4-Methylguaiacol has been proven to be a potent natural antioxidant in Baijiu [37]. 4-Ethylguaiacol and 4-ethylphenol had significant differences between the two groups of low-grade (F00, F10) and high-grade (F20, F30). *β*-Damascenone, whose aroma is always described as honey, floral and fruity, was a very important aroma substance in LFB [9,10], and its concentration was higher in F10 and F30. The concentrations of 2-pentylfuran, dimethyl trisulfide and naphthalene were relatively higher in F30, while 1,1-diethoxyethane and furfural had higher concentrations in F10. Dimethyl trisulfide, one of the most important sulfides in LFB [10], exhibits cabbage and spicy notes and has been proven to play an indispensable role in promoting the aroma quality of CYW [13].

### 3.3. OAVs

As the samples were significantly concentrated before GC-O analysis, the osme values could not indicate the final impact of these aroma-active compounds on the overall aroma in authentic liquors. So, OAVs were calculated to further confirm the contributions of these compounds [38]. The results are listed in Table 3. There were 28 aroma compounds with OAVs >1 simultaneously in four samples, which indicated that these odorants might contribute to the characteristic aroma. The most important aroma compounds (OAVs > 100) in all samples were 3-methylbutanal, *β*-damascenone and ethyl octanoate. Dimethyl trisulfide, ethyl hexanoate, ethyl acetate, 3-methylbutyl acetate, 1,1-diethoxyethane, ethyl butanoate, hexanal, ethyl 2-methylpropanoate, ethyl 3-methylbutanoate, ethyl lactate and ethyl pentanoate could also be significant according to their high OAVs (>10). Interestingly, recent studies have found that ethyl hexanoate, with the aroma of pineapple, also makes great contributions to the pungency and sweetness perception of Baijiu [35,39]. Compounds, including 1-octen-3-ol, ethyl 2-methylbutanoate, 2-methylpropanol, butanoic acid, acetic acid, pentanoic acid, 1-propanol, 1-butanol, guaiacol, phenylacetaldehyde, *β*-ionone, hexanoic acid, 3-methylbutanoic acid, 3-methylbutanol were other aroma contributors (OAVs > 1). The OAVs of nonanal and ethyl dodecanoate were greater than 1 only in F30. 

To visually compare the differences in those aroma-active compounds with OAVs > 1, the histogram of their concentrations in the four different grades of FJ is shown in Appendix A. Esters made a great contribution to the pleasant fruity and floral aroma of LFB. In F30, there were 12 esters considered as the main contributors of aroma, including ethyl octanoate, ethyl hexanoate, ethyl acetate, 3-methylbutyl acetate, ethyl butanoate, ethyl 2-methylpropanoate, ethyl 3-methylbutanoate, ethyl lactate, ethyl pentanoate, ethyl 2-methylbutanoate, ethyl decanoate, and ethyl dodecanoate. Aroma compounds, whose concentrations rose with the increase of the grade, may be the reason for the differences in aroma quality. Among these esters, the higher the sample grade, the higher the concentrations of ethyl butanoate, ethyl 2-methylbutanoate, ethyl 3-methylbutanoate, 3-methylbutyl acetate, ethyl hexanoate, ethyl octanoate, ethyl decanoate, and ethyl dodecanoate. Although the contents of acids and alcohols were generally high in Baijiu samples, their OAVs were almost all less than 10 in this paper due to their high threshold. Among acids and alcohols, the higher the sample grade, the higher the concentrations of 1-butanol, 1-hexanol, acetic acid, and hexanoic acid. However, some substances, such as *β*-damascenone and dimethyl trisulfide, whose contents were at a trace level, had a greater contribution to the aroma of LFB due to their low threshold [9,10].

### 3.4. Sensory Evaluation

#### 3.4.1. TDS

TDS is an effective tool to characterize the differences between samples during food or beverage consumption, which has been used to distinguish coffee added with different sweeteners [42], brandy with different aging times [17] and chocolate bars with different cocoa contents [43]. In this study, TDS curves showed the evolution of the retronasal aroma dominance rate over time after spitting (Figure 2A). The amount of dominant retronasal aroma attributes (the attribute which has the highest dominance rate) varied between samples. A dominant retronasal aroma attribute was defined as the most noticeable retronasal aroma attribute at a given time. It should be understood mostly as the new aroma popping up. Therefore, the dominant retronasal aroma attribute is not necessarily the one with the highest intensity. Regarding the TDS curves of F00, grain, alcoholic, fruity and roasted aromas were considered the dominant retronasal aroma attributes. In F10, alcohol was the first dominant aroma, followed by acidic, fruity, grassy, and fruity aromas. F20 had seven dominant retronasal aroma attributes, which was the most. They were alcoholic, grain, fruity (appeared twice), floral, roasted, grassy and earthy. The dominant retronasal aroma attributes in F30 were alcoholic, acidic, fruity, roasted, grassy and earthy.

TDS curves revealed some differences in the duration of the dominant retronasal aroma attributes between four kinds of differently graded commercial liquors. Figure 2A showed that the low-grade FJ (F00 and F10) had a short duration of retronasal aroma. Basically, no attributes were beyond the “significance level” after 50 s. After 70 s, panelists could hardly feel any retronasal aroma. However, high-grade FJ (F20 and F30) had a long and abundant perception of retronasal aroma. This may be related to the aging time of their base liquor. F00 had the longest dominant time of the grain and roasted aroma, while F30 had the longest one of acidic, fruity, grassy and earthy. TDS parameters, including DR-max (Maximum Dominance rate), T-max (Time for DR-max) and T-90%max (Time interval in which dominance rate is ≥90% of DR-max), effectively distinguished four kinds of FJ. The PCA score plot (Figure 2B) highlighted the similarity between F20 and F30. F00 had a higher maximum dominant rate of fruity, grain and roasted, while F20 and F30 had a higher maximum dominant rate of grassy, floral and earthy. Thus it could be seen that TDS can well characterize the sensory differences between different grades of LFB during consumption.

#### 3.4.2. QDA

TDS effectively characterized the differences in retronasal aromas on different grades of FJ. In addition to the retronasal aroma, the orthonasal aroma is also an important aspect of Baijiu aroma quality. However, TDS cannot be used for the static sensory analysis and the characterization of the aroma attribute intensity. Therefore, QDA was selected to compare the orthonasal aroma differences of the samples. The retronasal aroma of roasted and earthy could hardly be perceived in QDA. In addition, “sweet“ was a new attribute that referred to the aroma of sweet honey rather than sweetness in the taste. The radar chart showed (Figure 3A) that the four different grades of FJ had significant differences in all aroma attributes. The aroma profiles of F20 and F30 were relatively similar. PCA score plot (Figure 3B) demonstrated that there were significant differences between high-grade and low-grade samples, which were located on two sides of the coordinate axis. F00 and F10 mainly presented the aroma of grain and acidic, while high-grade F20 and F30 mainly exhibited the pleasant aroma of floral, fruity, sweet and grassy. The differences in sensory perception may be caused by the differences in aroma characteristics of base liquor used for blending. Some studies have proven that the aroma profile of Baijiu changes significantly during aging [11,12].

### 3.5. Prediction of the Potential Aroma Quality-Markers

Multivariate analysis has been widely used to evaluate the characteristics and differences of the samples [15,44,45]. In order to predict the potential marker compounds, the associations between the orthonasal aroma attributes (Y variables, n = 7) and the aroma compounds with OAVs >1 among the four FJ samples (X variables, n = 28) were analyzed by PLSR analysis [13]. The correlation loading plot of the aroma compounds and orthonasal aroma attributes among the four different grades of FJ are presented in Figure 4A. Dimension 1 explained 53.8% of the predictor variables (the concentrations of the aroma compounds) and 83.3% of the response variance (the intensities of the orthonasal aroma), while dimension 2 explained 82.6% of the predictor variables and 96.8% of the response variance. A majority of the aroma attributes and the aroma-active compounds were correlated to dimension 1 and had a positive contribution to the aroma of F20 and F30. The PLSR model showed that most esters, such as ethyl hexanoate, ethyl 2-methylbutanoate, ethyl 3-methylbutanoate, ethyl butanoate, and 3-methylbutyl acetate, were associated with the sweet, floral, fruity and grassy aroma. 2-Methylpropanol, 3-methylbutanol and 1-butanol were correlated with the sweet aroma. Acetic acid was considered an active contributor to fruity and floral aroma perception. The grassy aroma had a close correlation with hexanoic acid. Among these 28 odorants with OAVs >1, 15 with variable importance in projection (VIP) values >1 (Figure 4B), including 3-methylbutyl acetate, 1-butanol, ethyl hexanoate, ethyl butanoate, ethyl octanoate, acetic acid, 3-methylbutanol, hexanoic acid, 1-octen-3-ol, phenylacetaldehyde, ethyl 2-methylpropanoate, ethyl 2-methylbutanoate, 3-methylbutanoic acid, ethyl 3-methylbutanoate, and 2-methylpropanol, were considered as the potential odorants causing aroma differences among the four different grades of FJ. However, a large number of samples and omission tests were still needed to identify the quality markers which cause the different quality grades of FJ.

## 4. Conclusions

In summary, a total of 88 aroma-active compounds were identified by LLE combined with GC-O-MS analysis in four different grades of FJ. Among them, six aroma compounds were identified as active aroma components in LFB for the first time, although they may make fewer contributions to the overall aroma because of their low osme values. GC-O analysis showed that high-grade FJ had more aroma compounds and higher aroma intensities. Seventy aroma compounds were quantified by three different methodologies. The contents of most aroma compounds in high-grade FJ were higher than those in low-grade FJ. There were 28 aroma compounds with OAVs >1 simultaneously in four samples, which indicated that these odorants might contribute to the characteristic aroma of FJ. TDS well characterized the aroma quality differences of four different grades of FJ during consumption and suggested that high-grade FJ had a rich, pleasant and lasting retronasal aroma perception. The QDA results showed that low-grade FJ mainly presented the aroma of grain and acidic, while high-grade FJ mainly exhibited the pleasant aroma of floral, fruity, sweet and grassy. The combination of TDS and QDA could more comprehensively characterize the sensory quality of LFB. PLSR analysis effectively distinguished 4 kinds of FJ and revealed the associations between the orthonasal aroma attributes and the aroma compounds with OAVs > 1. 3-Methylbutyl acetate, 1-butanol, ethyl hexanoate, ethyl butanoate, ethyl octanoate, acetic acid, 3-methylbutanol, hexanoic acid, 1-octen-3-ol, phenylacetaldehyde, ethyl 2-methylpropanoate, ethyl 2-methylbutanoate, 3-methylbutanoic acid, ethyl 3-methylbutanoate, and 2-methylpropanol with their VIP values > 1 were considered as the odorants causing aroma differences among the four different grades of FJ. However, a large number of samples and omission tests were still needed to identify the quality markers which cause the different quality grades of FJ.

A better understanding of the different grades of LFB will help improve the flavor quality and processing technology and increase the yield of high-grade Baijiu products. This study paid attention to the aroma quality of different grades of LFB during consumption for the first time. During the consumption of alcoholic beverages, retronasal aroma perception plays a crucial role in determining the final aroma characteristics and significantly affects consumer preferences. In addition, future research should be focused on the dynamic analysis of the retronasal aroma compounds and the exploration of the relationship between retronasal aroma perception and release.

## Figures and Tables

**Figure 1 foods-12-01238-f001:**
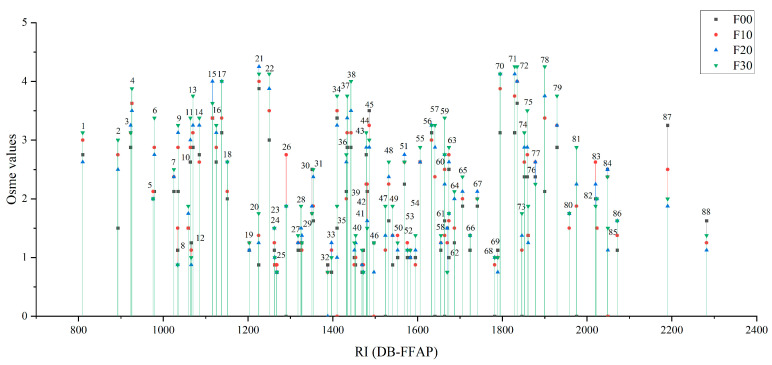
Osme value comparison of four different grades of FJ. Notations: The number on the peak corresponds to Table 1, the abscissa represents the retention index of the compounds on the DB-FFAP chromatographic column, and the ordinate represents the aroma intensity of the compounds.

**Figure 2 foods-12-01238-f002:**
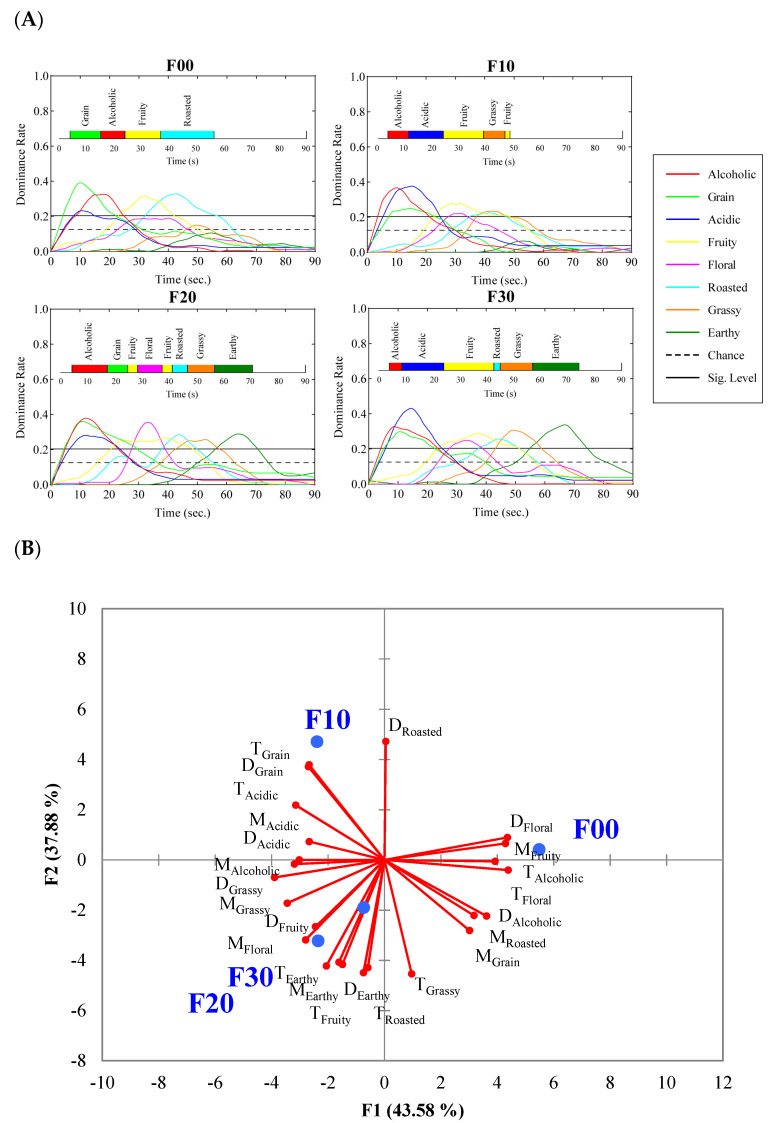
(**A**) TDS curves of four different grades of FJ. Notations: Chance level is defined as the dominance rate that an attribute can be obtained by chance, and its value is equal to the reciprocal of the attribute number. The significance level represents the minimum value the dominance rate should equal to be considered significantly higher than the chance level. It is calculated using the confidence interval of a binomial proportion based on a normal approximation. (**B**) PCA score plot based on the parameters that summarize the TDS curves of four samples. M = maximum dominance rate, T = time of the maximum dominance, D = Time interval when the dominance rate is ≥90% of the maximum dominance rate.

**Figure 3 foods-12-01238-f003:**
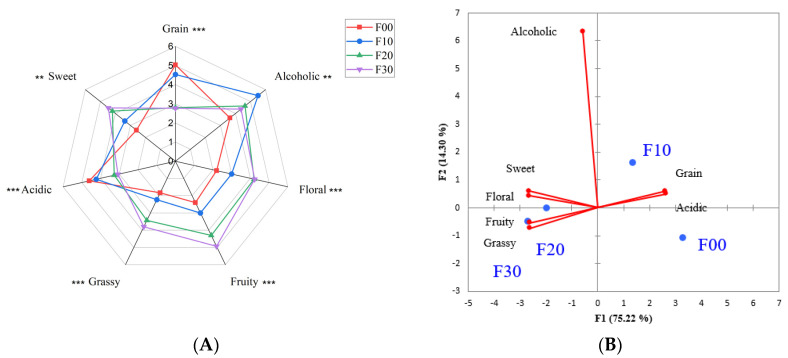
(**A**) Aroma profile of the mean sensory scores of the four different grades of FJ. Notations: “ ** ” and “ *** ” indicate significance at *p* < 0.01 and *p* < 0.001, respectively. (**B**) PCA score plot based on the sensory scores.

**Figure 4 foods-12-01238-f004:**
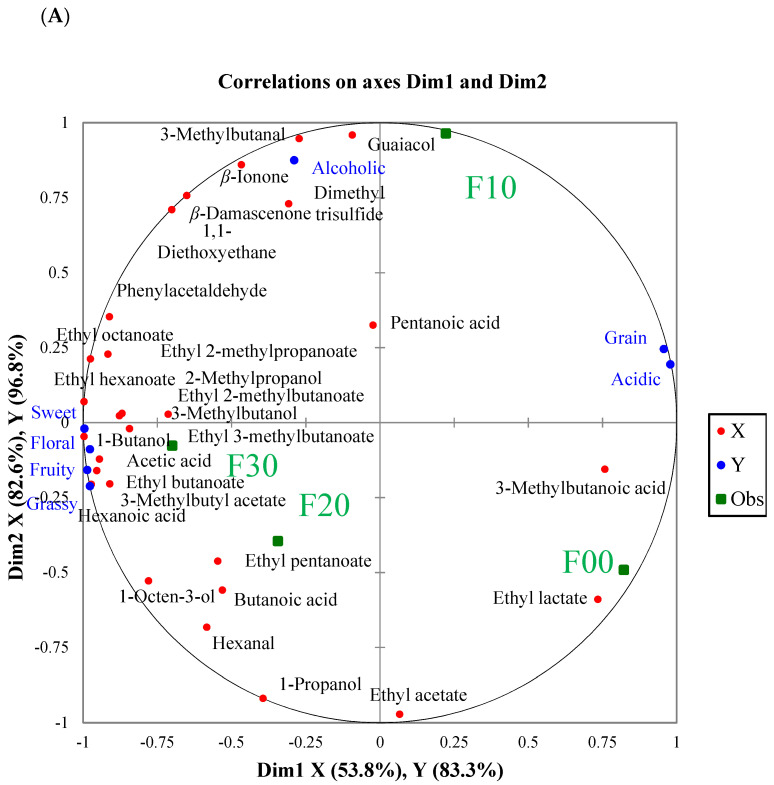
(**A**) Correlation loading plot between the aroma compounds with OAVs >1 (X) and orthonasal aroma attributes (Y) among the four different grades of FJ (Obs). (**B**) Variable importance for the projection (VIP) values for 28 aroma compounds (OAVs > 1).

**Table 1 foods-12-01238-t001:** Aroma compounds identified by GC-O-MS in four different grades of FJ.

No.	Aroma Compound	Descriptor ^A^	CAS	Basis of ID ^B^	RI ^C^		Osme Values ^D^			
					DB-FFAP	HP-5	F00	F10	F20	F30
1	2-Methylpropanal	Grass	78-84-2	MS, Aroma, RI, S	810	550	2.75 ^a^	3.00 ^a^	2.63 ^a^	3.13 ^a^
2	1,1-Diethoxyethane	Fruity	105-57-7	MS, Aroma, RI, S	893	725	1.50 ^b^	2.75 ^a^	2.50 ^a^	3.00 ^a^
3	3-Methylbutanal	Malty	590-86-3	MS, Aroma, RI, S	923	625	2.88 ^a^	3.13 ^a^	3.25 ^a^	3.13 ^a^
4	Ethyl acetate	Pineapple	141-78-6	MS, Aroma, RI, S	926	608	3.63 ^a^	3.63 ^a^	3.50 ^a^	3.88 ^a^
5	Ethyl propanoate	Banana	105-37-3	MS, Aroma, RI, S	976	711	2.00 ^a^	2.13 ^a^	2.00 ^a^	2.00 ^a^
6	Ethyl 2-methylpropanoate	Fruity	97-62-1	MS, Aroma, RI, S	979	756	2.13 ^b^	2.88 ^a^	2.75 ^ab^	3.38 ^a^
7	2-Methylpropyl acetate	Fruity	110-19-0	MS, Aroma, RI, S	1025	772	2.13 ^a^	2.38 ^a^	2.38 ^a^	2.50 ^a^
8	2-Butanol	Fruity	78-92-2	MS, Aroma, RI, S	1034	-	1.13 ^a^	1.50 ^a^	0.88 ^a^	0.88 ^a^
9	Ethyl butanoate	Pineapple	105-54-4	MS, Aroma, RI, S	1035	815	2.13 ^b^	2.88 ^ab^	3.13 ^a^	3.25 ^a^
10	1-Propanol	Alcoholic	71-23-8	MS, Aroma, RI, S	1059	-	1.50 ^a^	1.50 ^a^	1.75 ^a^	1.88 ^a^
11	Ethyl 2-methylbutanoate	Berry	7452-79-1	MS, Aroma, RI, S	1064	843	2.63 ^b^	2.88 ^ab^	3.00 ^ab^	3.38 ^a^
12	Dimethyl disulfide	Onion, cabbage	624-92-0	MS, Aroma, RI, S	1066	-	1.25 ^a^	1.13 ^a^	0.88 ^a^	1.00 ^a^
13	Ethyl 3-methylbutanoate	Apple	108-64-5	MS, Aroma, RI, S	1070	856	2.88 ^a^	3.13 ^a^	3.25 ^a^	3.75 ^a^
14	Hexanal	Grassy, green	66-25-1	MS, Aroma, RI, S	1085	808	2.75 ^ab^	2.63 ^b^	3.25 ^ab^	3.38 ^a^
15	2-Methylpropanol	Malty	78-83-1	MS, Aroma, RI, S	1116	623	3.38 ^a^	3.38 ^a^	4.00 ^a^	3.63 ^a^
16	1-Butanol	Alcoholic, solvent	71-36-3	MS, Aroma, RI, S	1125	667	2.63 ^a^	2.88 ^a^	3.13 ^a^	3.25 ^a^
17	3-Methylbutyl acetate	Fruity	123-92-2	MS, Aroma, RI, S	1138	856	3.13 ^b^	3.38 ^ab^	4.00 ^a^	4.00 ^a^
18	Ethyl pentanoate	Apple	539-82-2	MS, Aroma, RI, S	1151	909	2.00 ^a^	2.13 ^a^	2.63 ^a^	2.63 ^a^
19	2-Heptanone	Soap	110-43-0	MS, Aroma, RI, S	1203	-	1.13 ^a^	1.25 ^a^	1.13 ^a^	1.25 ^a^
20	2-Pentylfuran	Green bean	3777-69-3	MS, Aroma, RI, S	1225	-	0.88 ^b^	1.38 ^ab^	1.25 ^ab^	1.75 ^a^
21	3-Methylbutanol	Malty	123-51-3	MS, Aroma, RI, S	1226	786	3.88 ^a^	4.00 ^a^	4.25 ^a^	4.13 ^a^
22	Ethyl hexanoate	Fruity	123-66-0	MS, Aroma, RI, S	1250	1020	3.00 ^b^	3.5 ^ab^	3.88 ^a^	4.13 ^a^
23	1-Pentanol	Balsamic	71-41-0	MS, Aroma, RI, S	1262	760	1.13 ^a^	1.25 ^a^	1.50 ^a^	1.50 ^a^
24	Hexyl acetate	Fruity	142-92-7	MS, Aroma, RI, S	1263	1008	0.88 ^a^	0.88 ^a^	1.00 ^a^	1.00 ^a^
25	3-Methylbutyl butanoate	Fruity	106-27-4	MS, Aroma, RI, S	1268	-	0.75 ^a^	0.88 ^a^	0.75 ^a^	0.75 ^a^
26	3-Hydroxy-2-butanone	Butter, cream	513-86-0	MS, Aroma, RI, S	1290	-	-	2.75 ^a^	1.88 ^b^	1.88 ^b^
27	2-Heptanol	Fruity	543-49-7	MS, Aroma, RI, S	1318	900	1.13 ^a^	1.25 ^a^	1.25 ^a^	1.38 ^a^
28	Ethyl heptanoate	Fruity	106-30-9	MS, Aroma, RI, S	1325	1103	1.13 ^b^	1.25 ^ab^	1.5 ^ab^	1.88 ^a^
29	2,6-Dimethylpyrazine	Roasted nut	108-50-9	Aroma, RI, S	1327	-	1.25 ^a^	1.13 ^a^	1.38 ^a^	1.25 ^a^
30	Ethyl lactate	Fruity	97-64-3	MS, Aroma, RI, S	1351	827	2.50 ^a^	1.75 ^a^	1.88 ^a^	1.75 ^a^
31	1-Hexanol	Grass	111-27-3	MS, Aroma, RI, S	1354	904	1.63 ^a^	1.88 ^a^	2.38 ^a^	2.50 ^a^
32	2-Nonanone	Floral	821-55-6	MS, Aroma, RI, S	1388	-	0.88 ^a^	0.75 ^a^	-	0.75 ^a^
33	3-Hexen-1-ol, (*Z*)-	Grass	928-96-1	MS, Aroma, RI, S	1397	-	0.75 ^a^	1.13 ^a^	1.25 ^a^	1.00 ^a^
34	Dimethyl trisulfide	Sulfur, cabbage	3658-80-8	MS, Aroma, RI, S	1410	975	3.38 ^a^	3.50 ^a^	3.25 ^a^	3.75 ^a^
35	Nonanal	Soapy	124-19-6	MS, Aroma, RI, S	1410	1120	1.50 ^ab^	-	1.00 ^b^	1.88 ^a^
36	Trimethylpyrazine	Roast, potato	14667-55-1	MS, Aroma, RI, S	1432	-	2.13 ^a^	2.00 ^a^	2.63 ^a^	2.75 ^a^
37	Acetic acid	Acidic, vinegar	64-19-7	MS, Aroma, RI, S	1434	605	2.88 ^b^	3.13 ^ab^	3.38 ^ab^	3.75 ^a^
38	Ethyl octanoate	Fruity	106-32-1	MS, Aroma, RI, S	1443	1199	2.88 ^b^	3.13 ^b^	3.50 ^ab^	4.00 ^a^
39	1-Heptanol	Alcoholic	111-70-6	MS, Aroma, RI, S	1451	980	1.00 ^a^	0.88 ^a^	1.13 ^a^	1.25 ^a^
40	3-Methylbutyl hexanoate	Fruity	2198-61-0	MS, Aroma, RI, S	1454	1263	0.88 ^a^	1.00 ^a^	1.25 ^a^	1.38 ^a^
41	Octyl acetate	Green	112-14-1	MS, Aroma, RI, S	1469	-	0.75 ^a^	0.88 ^a^	0.88 ^a^	1.13 ^a^
42	2-Furaldehyde diethyl acetal	Earthy	13529-27-6	MS, Aroma, RI, S	1472	-	1.13 ^a^	0.88 ^a^	0.75 ^a^	0.75 ^a^
43	1-Octen-3-ol	Mushroom	3391-86-4	MS, Aroma, RI, S	1479	982	2.75 ^ab^	2.25 ^b^	2.88 ^ab^	3.13 ^a^
44	Furfural	Sweet, almond	98-01-1	MS, Aroma, RI, S	1481	842	2.13 ^ab^	2.25 ^a^	1.63 ^ab^	1.50 ^b^
45	Tetramethylpyrazine	Nutty	1124-11-4	MS, Aroma, RI, S	1486	1094	3.50 ^a^	3.25 ^a^	2.88 ^a^	3.00 ^a^
46	Decanal	Oily	112-31-2	MS, Aroma, RI, S	1497	1210	1.25 ^a^	-	0.75 ^a^	1.25 ^a^
47	Benzaldehyde	Almond, burnt sugar	100-52-7	MS, Aroma, RI, S	1524	980	-	1.13 ^a^	1.38 ^a^	1.88 ^a^
48	Ethyl nonanoate	Fruity	123-29-5	MS, Aroma, RI, S	1532	1293	1.63 ^b^	2.25 ^ab^	2.38 ^a^	2.63 ^a^
49	Propanoic acid	Vinegar	79-09-4	MS, Aroma, RI, S	1541	-	0.88 ^b^	1.38 ^ab^	1.38 ^ab^	1.88 ^a^
50	1-Octanol	Fruity	111-87-5	MS, Aroma, RI, S	1553	1096	1.00 ^a^	1.38 ^a^	1.13 ^a^	1.25 ^a^
51	2-Methylpropanoic acid	Sweaty, acidic	79-31-2	MS, Aroma, RI, S	1569	783	2.25 ^a^	2.63 ^a^	2.75 ^a^	2.63 ^a^
52	2,3-Butanediol	Fruit, onion	513-85-9	MS, Aroma, RI, S	1576	806	1.00 ^a^	1.25 ^a^	1.13 ^a^	1.13 ^a^
53	Diethyl malonate	Apple	105-53-3	MS, Aroma, RI, S	1583	-	1.00 ^a^	1.13 ^a^	1.00 ^a^	1.13 ^a^
54	5-Methyl furfural	Caramel, burnt sugar	620-02-0	MS, Aroma, RI, S	1595	-	1.00 ^a^	0.88 ^a^	1.13 ^a^	1.38 ^a^
55	(*E*,*Z*)-2,6-nonadienal	Green, cucumber	557-48-2	Aroma, RI, S	1606	1230	2.63 ^a^	2.63 ^a^	2.63 ^a^	2.88 ^a^
56	Butanoic acid	Sweaty, acidic	107-92-6	MS, Aroma, RI, S	1633	808	3.13 ^a^	3.00 ^a^	3.25 ^a^	3.25 ^a^
57	Ethyl decanoate	Fruity	110-38-3	MS, Aroma, RI, S	1641	1393	-	2.38 ^b^	2.88 ^ab^	3.25 ^a^
58	1-Nonanol	Fat, green	143-08-8	MS, Aroma, RI, S	1655	1163	1.13 ^a^	1.25 ^a^	1.25 ^a^	1.38 ^a^
59	Phenylacetaldehyde	Floral	122-78-1	MS, Aroma, RI, S	1664	1056	1.63 ^c^	2.50 ^b^	2.75 ^ab^	3.38 ^a^
60	Ethyl benzoate	Fruity	93-89-0	MS, Aroma, RI, S	1664	1170	-	1.38 ^b^	2.25 ^a^	2.75 ^a^
61	2-Furanmethanol	Burnt	98-00-0	MS, Aroma, RI, S	1670	935	1.50 ^a^	1.25 ^a^	1.50 ^a^	0.75 ^a^
62	2-Methylbutanoic acid	Sweaty, acidic	116-53-0	MS, Aroma, RI, S	1674	942	1.00 ^a^	1.63 ^a^	1.75 ^a^	1.75 ^a^
63	3-Methylbutanoic acid	Sweaty, acidic	503-74-2	MS, Aroma, RI, S	1674	842	2.50 ^a^	2.75 ^a^	2.63 ^a^	2.88 ^a^
64	Diethyl butanedioate	Fruity	123-25-1	MS, Aroma, RI, S	1687	1188	1.25 ^b^	1.50 ^ab^	2.00 ^ab^	2.13 ^a^
65	2-Thiophenecarboxaldehyde	Sulfur	98-03-3	Aroma, RI, S	1706	-	1.88 ^a^	2.00 ^a^	2.13 ^a^	2.38 ^a^
66	γ-Hexalactone	Coumarin, sweet	695-06-7	MS, Aroma, RI, S	1724	-	1.13 ^a^	1.38 ^a^	1.38 ^a^	1.38 ^a^
67	Pentanoic acid	Sweaty, rancid	109-52-4	MS, Aroma, RI, S	1741	-	1.88 ^a^	2.00 ^a^	2.13 ^a^	2.00 ^a^
68	Naphthalene	Mothball-like	91-20-3	MS, Aroma, RI, S	1782	-	-	0.88 ^a^	1.00 ^a^	1.00 ^a^
69	2(5H)-Furanone	Buttery	497-23-4	MS, Aroma, RI, S	1789	961	1.13 ^a^	1.00 ^a^	0.75 ^a^	1.00 ^a^
70	Ethyl phenylacetate	Rosy, honey	101-97-3	MS, Aroma, RI, S	1795	1252	3.13 ^b^	3.88 ^ab^	4.13 ^a^	4.13 ^a^
71	2-Phenylethyl acetate	Floral	103-45-7	MS, Aroma, RI, S	1829	1274	3.13 ^b^	3.75 ^ab^	4.13 ^a^	4.25 ^a^
72	*β*-Damascenone	Floral, honey	23726-93-4	MS, Aroma, RI, S	1835	1375	3.63 ^a^	4.00 ^a^	4.00 ^a^	4.25 ^a^
73	Ethyl dodecanoate	Leaf	106-33-2	MS, Aroma, RI, S	1846	1585	-	1.13 ^a^	1.38 ^a^	1.75 ^a^
74	Hexanoic acid	Sweaty	142-62-1	MS, Aroma, RI, S	1852	984	2.38 ^b^	2.63 ^ab^	2.88 ^ab^	3.13 ^a^
75	Geosmin	Earth	19700-21-1	MS, Aroma, RI, S	1859	1450	2.38 ^b^	2.75 ^b^	2.88 ^b^	3.50 ^a^
76	Geranylacetone	Floral	3796-70-1	MS, Aroma, RI, S	1861	1457	1.38 ^a^	1.38 ^a^	1.25 ^a^	1.88 ^a^
77	Guaiacol	Clove	90-05-1	MS, Aroma, RI, S	1878	1096	2.38 ^a^	2.63 ^a^	2.63 ^a^	2.25 ^a^
78	Ethyl 3-phenylpropanoate	Floral	2021-28-5	MS, Aroma, RI, S	1900	1346	2.13 ^c^	3.38 ^b^	3.75 ^ab^	4.25 ^a^
79	2-Phenylethanol	Rosy, honey	60-12-8	MS, Aroma, RI, S	1929	1122	2.88 ^b^	3.25 ^ab^	3.25 ^ab^	3.75 ^a^
80	*β*-Ionone	Floral	79-77-6	MS, Aroma, RI, S	1958	1470	1.75 ^a^	1.50 ^a^	1.75 ^a^	1.75 ^a^
81	4-Methylguaiacol	Smoky	93-51-6	MS, Aroma, RI, S	1975	1195	-	1.88 ^b^	2.25 ^ab^	2.88 ^a^
82	Phenol	Medicinal	108-95-2	MS, Aroma, RI, S	2020	993	2.00 ^a^	2.63 ^a^	2.25 ^a^	1.88 ^a^
83	*γ*-Nonalactone	Coconut, peach	104-61-0	Aroma, RI, S	2023	-	1.50 ^a^	1.5 ^a^	2.00 ^a^	2.00 ^a^
84	4-Ethylguaiacol	Clove	2785-89-9	MS, Aroma, RI, S	2048	1276	2.50 ^a^	2.38 ^a^	2.50 ^a^	2.38 ^a^
85	Ethyl tetradecanoate	Coconut	124-06-1	MS, Aroma, RI, S	2049	1793	-	-	1.13 ^a^	1.50 ^a^
86	Octanoic acid	Cheesy	124-07-2	MS, Aroma, RI, S	2071	1287	1.13 ^a^	1.38 ^a^	1.63 ^a^	1.63 ^a^
87	4-Ethylphenol	Animal	123-07-9	MS, Aroma, RI, S	2190	-	3.25 ^a^	2.50 ^b^	1.88 ^c^	2.00 ^bc^
88	Decanoic acid	Sweaty	334-48-5	MS, Aroma, RI, S	2282	1378	1.63 ^a^	1.25 ^a^	1.13 ^a^	1.38 ^a^

^A^ Odor quality perceived at the sniffing port. ^B^ Identification based on MS (mass spectrometry), aroma (odor description by comparison to the reference standards by GC-O), RI (retention index) and S (standards). ^C^ Retention indices determined by GC-MS on two different stationary phases (DB-FFAP and DB-5). ^D^ The average value of four panelists. Different superscript letters indicate significant statistical differences (*p* < 0.05) among samples. -, not detected.

**Table 2 foods-12-01238-t002:** Concentrations of aroma compounds in four different grades of FJ.

No.	Aroma Compound	Concentration (μg/L)			
		F00	F10	F20	F30
Esters					
4	Ethyl acetate *	1282.25 ± 35.69 ^a^	1006.35 ± 25.97 ^c^	1205.99 ± 17.75 ^b^	1209.86 ± 33.36 ^b^
5	Ethyl propanoate	5824.68 ± 489.78 ^a^	6142.42 ± 892.68 ^a^	4922.67 ± 318.86 ^ab^	4137.84 ± 135.58 ^b^
6	Ethyl 2-methylpropanoate	966.65 ± 34.52 ^c^	1252.69 ± 51.26 ^b^	1243.91 ± 34.23 ^b^	1595.64 ± 75.59 ^a^
7	2-Methylpropyl acetate	733.23 ± 66.56 ^b^	784.76 ± 49.95 ^ab^	837.95 ± 41.63 ^ab^	892.57 ± 54.36 ^a^
9	Ethyl butanoate	1943.70 ± 215.88 ^b^	2069.28 ± 190.70 ^b^	2385.55 ± 132.67 ^ab^	2759.18 ± 196.80 ^a^
11	Ethyl 2-methylbutanoate	102.60 ± 10.20 ^c^	124.40 ± 7.72 ^bc^	132.31 ± 4.04 ^b^	198.22 ± 14.72 ^a^
13	Ethyl 3-methylbutanoate	107.64 ± 11.05 ^c^	127.67 ± 7.81 ^bc^	138.02 ± 5.39 ^b^	230.55 ± 18.55 ^a^
17	3-Methylbutyl acetate	3040.63 ± 419.90 ^b^	3298.69 ± 349.05 ^ab^	4008.56 ± 249.74 ^a^	4102.69 ± 327.75 ^a^
18	Ethyl pentanoate	293.59 ± 5.56 ^c^	287.05 ± 11.53 ^c^	513.16 ± 32.43 ^a^	348.26 ± 14.69 ^b^
22	Ethyl hexanoate	4105.70 ± 52.29 ^c^	4754.95 ± 72.99 ^b^	5193.83 ± 258.16 ^a^	5526.09 ± 93.14 ^a^
24	Hexyl acetate	31.54 ± 0.60 ^b^	33.05 ± 0.15 ^ab^	37.43 ± 3.86 ^a^	33.50 ± 0.61 ^ab^
25	3-Methylbutyl butanoate	6.80 ± 0.11 ^ab^	7.59 ± 0.91 ^a^	7.44 ± 0.97 ^ab^	5.81 ± 0.36 ^b^
28	Ethyl heptanoate	152.20 ± 1.36 ^c^	171.75 ± 1.70 ^b^	175.94 ± 7.63 ^b^	189.25 ± 4.22 ^a^
30	Ethyl lactate *	1710.94 ± 58.10 ^a^	1446.91 ± 191.61 ^ab^	1589.86 ± 51.69 ^ab^	1366.27 ± 143.05 ^b^
38	Ethyl octanoate	2470.72 ± 64.20 ^d^	3402.79 ± 100.54 ^c^	3647.25 ± 34.21 ^b^	4170.69 ± 73.54 ^a^
40	3-Methylbutyl hexanoate	3.84 ± 0.16 ^d^	6.58 ± 0.12 ^c^	7.46 ± 0.08 ^b^	8.46 ± 0.06 ^a^
41	Octyl acetate	1.49 ± 0.04 ^c^	3.26 ± 0.05 ^a^	2.62 ± 0.20 ^b^	2.69 ± 0.01 ^b^
48	Ethyl nonanoate	85.28 ± 3.40 ^c^	152.06 ± 8.65 ^b^	160.07 ± 0.86 ^b^	219.89 ± 5.22 ^a^
57	Ethyl decanoate	164.75 ± 38.84 ^d^	1111.61 ± 68.95 ^c^	1407.43 ± 34.29 ^b^	2141.21 ± 98.71 ^a^
60	Ethyl benzoate	8.12 ± 0.80 ^d^	73.61 ± 8.45 ^c^	117.80 ± 3.76 ^b^	241.36 ± 7.19 ^a^
64	Diethyl butanedioate *	13.87 ± 0.85 ^c^	19.76 ± 1.31 ^ab^	23.66 ± 3.85 ^a^	16.91 ± 0.75 ^bc^
70	Ethyl phenylacetate	48.57 ± 3.06 ^b^	86.47 ± 7.37 ^a^	93.20 ± 11.43 ^a^	98.17 ± 4.68 ^a^
71	2-Phenylethyl acetate	95.07 ± 6.64 ^c^	185.70 ± 10.66 ^b^	223.70 ± 26.14 ^a^	228.95 ± 11.14 ^a^
73	Ethyl dodecanoate	41.53 ± 3.18 ^d^	192.83 ± 17.83 ^c^	310.44 ± 30.74 ^b^	498.46 ± 83.15 ^a^
78	Ethyl 3-phenylpropanoate	23.45 ± 1.51 ^d^	59.31 ± 4.95 ^c^	83.11 ± 8.79 ^b^	109.18 ± 6.12 ^a^
85	Ethyl tetradecanoate	13.07 ± 1.01 ^c^	37.94 ± 3.69 ^c^	66.60 ± 11.04 ^b^	148.83 ± 20.91 ^a^
Alcohols					
10	1-Propanol *	147.18 ± 2.86 ^a^	130.54 ± 5.46 ^b^	153.10 ± 3.00 ^a^	150.41 ± 2.29 ^a^
15	2-Methylpropanol *	105.36 ± 3.33 ^b^	117.60 ± 4.63 ^ab^	131.24 ± 9.78 ^a^	118.09 ± 1.83 ^ab^
16	1-Butanol	6836.51 ± 547.73 ^b^	7472.33 ± 273.04 ^ab^	8209.62 ± 990.77 ^ab^	8754.91 ± 262.29 ^a^
21	3-Methylbutanol *	210.58 ± 20.30 ^b^	235.82 ± 11.48 ^ab^	261.64 ± 17.46 ^a^	247.05 ± 7.30 ^a^
23	1-Pentanol	831.21 ± 48.90 ^c^	892.30 ± 52.45 ^bc^	1026.70 ± 84.01 ^ab^	1101.64 ± 110.24 ^a^
27	2-Heptanol	52.60 ± 1.04 ^d^	75.00 ± 1.08 ^b^	67.55 ± 1.15 ^c^	87.28 ± 1.94 ^a^
31	1-Hexanol	4683.95 ± 143.41 ^c^	5318.54 ± 111.35 ^bc^	6142.75 ± 589.31 ^ab^	6419.92 ± 371.13 ^a^
39	1-Heptanol	172.05 ± 8.02 ^b^	165.02 ± 4.77 ^b^	191.46 ± 12.92 ^a^	203.81 ± 3.33 ^a^
43	1-Octen-3-ol	62.85 ± 4.24 ^c^	47.38 ± 4.02 ^d^	79.35 ± 4.54 ^b^	101.39 ± 4.70 ^a^
50	1-Octanol	202.82 ± 11.58 ^c^	329.4 ± 20.87 ^a^	258.02 ± 3.07 ^b^	318.94 ± 21.73 ^a^
58	1-Nonanol	138.38 ± 1.88 ^b^	103.63 ± 4.05 ^c^	129.24 ± 10.21 ^b^	173.47 ± 7.79 ^a^
79	2-Phenylethanol	3627.27 ± 485.36 ^b^	5229.01 ± 690.47 ^a^	4995.13 ± 230.03 ^a^	5948.54 ± 460.99 ^a^
Acids					
37	Acetic acid *	464.55 ± 24.48 ^c^	497.28 ± 10.93 ^c^	558.76 ± 33.68 ^b^	655.99 ± 27.11 ^a^
49	Propanoic acid	5551.89 ± 30.24 ^b^	5507.75 ± 36.97 ^b^	5590.53 ± 41.86 ^b^	5765.54 ± 54.67 ^a^
51	2-Methylpropanoic acid	1564.50 ± 6.06 ^c^	1689.79 ± 36.05 ^b^	1748.42 ± 9.81 ^a^	1671.23 ± 12.23 ^b^
56	Butanoic acid	3233.10 ± 34.30 ^a^	3203.36 ± 14.03 ^a^	3409.23 ± 215.26 ^a^	3275.12 ± 8.66 ^a^
62	2-Methylbutanoic acid	632.55 ± 9.76 ^b^	656.37 ± 8.79 ^ab^	656.00 ± 12.06 ^ab^	682.52 ± 16.15 ^a^
63	3-Methylbutanoic acid	1255.95 ± 18.10 ^a^	1227.21 ± 10.64 ^b^	1207.21 ± 8.06 ^b^	1227.39 ± 6.40 ^b^
67	Pentanoic acid	1097.81 ± 7.71 ^a^	1102.06 ± 12.26 ^a^	1103.19 ± 14.53 ^a^	1096.73 ± 3.32 ^a^
74	Hexanoic acid	3883.36 ± 32.3 ^c^	3906.36 ± 25.35 ^c^	4039.82 ± 85.68 ^b^	4242.95 ± 51.26 ^a^
86	Octanoic acid	1758.77 ± 15.70 ^a^	1764.65 ± 24.35 ^a^	1824.91 ± 860.81 ^a^	1915.14 ± 56.46 ^a^
88	Decanoic acid	1082.33 ± 13.85 ^a^	1054.89 ± 2.52 ^a^	1051.03 ± 495.46 ^a^	1055.34 ± 497.50 ^a^
Aldehydes					
1	2-Methylpropanal	795.96 ± 40.60 ^a^	870.65 ± 45.39 ^a^	806.57 ± 59.97 ^a^	899.92 ± 79.19 ^a^
3	3-Methylbutanal *	10.12 ± 0.49 ^a^	11.41 ± 0.74 ^a^	10.31 ± 1.43 ^a^	10.89 ± 0.62 ^a^
14	Hexanal	521.66 ± 20.64 ^bc^	426.41 ± 26.08 ^c^	756.51 ± 48.89 ^a^	610.01 ± 62.85 ^b^
35	Nonanal	97.55 ± 6.93 ^b^	71.86 ± 12.35 ^c^	75.92 ± 13.27 ^bc^	139.25 ± 3.38 ^a^
46	Decanal	15.05 ± 2.03 ^a^	7.45 ± 1.93 ^b^	7.72 ± 0.51 ^b^	18.04 ± 4.63 ^a^
47	Benzaldehyde	11.69 ± 2.12 ^c^	34.53 ± 2.41 ^b^	32.61 ± 2.93 ^b^	85.34 ± 6.65 ^a^
59	Phenylacetaldehyde	476.50 ± 4.80 ^c^	1357.20 ± 67.64 ^b^	1255.01 ± 151.54 ^b^	1949.86 ± 156.87 ^a^
Ketones					
19	2-Heptanone	34.07 ± 2.79 ^a^	30.46 ± 1.06 ^a^	28.65 ± 3.21 ^a^	32.60 ± 2.81 ^a^
32	2-Nonanone	10.32 ± 0.81 ^a^	7.53 ± 1.19 ^b^	4.08 ± 1.33 ^c^	6.66 ± 0.71 ^b^
Phenols					
77	Guaiacol	33.04 ± 1.79 ^c^	71.45 ± 6.29 ^a^	30.12 ± 2.99 ^c^	51.78 ± 2.87 ^b^
81	4-Methylguaiacol	19.69 ± 0.75 ^d^	176.53 ± 36.23 ^b^	119.89 ± 17.64 ^c^	240.15 ± 14.17 ^a^
82	Phenol	603.54 ± 15 ^b^	811.84 ± 73.58 ^a^	667.88 ± 38.5 ^b^	580.00 ± 43.96 ^b^
84	4-Ethylguaiacol	18.80 ± 3.02 ^b^	20.83 ± 0.27 ^b^	26.64 ± 2.14 ^a^	29.28 ± 2.38 ^a^
87	4-Ethylphenol	44.70 ± 7.25 ^a^	43.76 ± 3.86 ^a^	24.36 ± 1.09 ^b^	29.77 ± 6.39 ^b^
Terpenes					
72	*β*-Damascenone	30.61 ± 1.61 ^c^	52.65 ± 2.78 ^a^	42.29 ± 1.34 ^b^	50.41 ± 5.82 ^a^
76	Geranylacetone	5.10 ± 0.31 ^b^	6.74 ± 0.40 ^b^	5.26 ± 0.96 ^b^	10.17 ± 1.27 ^a^
80	*β*-Ionone	2.20 ± 0.12 ^a^	2.75 ± 0.44 ^a^	2.35 ± 0.14 ^a^	2.64 ± 0.19 ^a^
Others					
2	1,1-Diethoxyethane *	51.09 ± 0.54 ^c^	65.43 ± 0.72 ^a^	60.28 ± 1.11 ^b^	64.09 ± 1.28 ^a^
20	2-Pentylfuran	40.48 ± 2.36 ^c^	75.20 ± 8.22 ^b^	68.34 ± 5.36 ^b^	150.31 ± 7.18 ^a^
34	Dimethyl trisulfide	33.57 ± 1.02 ^ab^	37.42 ± 1.97 ^a^	32.53 ± 0.89 ^b^	37.60 ± 2.55 ^a^
44	Furfural	7186.48 ± 203.23 ^b^	7894.88 ± 185.57 ^a^	6849.83 ± 304.94 ^b^	6690.93 ± 399.23 ^b^
68	Naphthalene	2.33 ± 0.11 ^c^	2.55 ± 0.13 ^c^	3.60 ± 0.18 ^b^	4.73 ± 0.46 ^a^

* The concentration was mg/L. Different superscript letters indicate significant statistical differences (*p* < 0.05) among samples.

**Table 3 foods-12-01238-t003:** The odor activity values (OAVs) of aroma compounds in four different grades of FJ.

No.	Aroma Compound	Threshold (μg/L)	OAVs			
			F00	F10	F20	F30
3	3-Methylbutanal	17 ^a^	595.53	671.19	606.69	640.78
72	*β*-Damascenone	0.12 ^a^	255.08	438.76	352.45	420.06
38	Ethyl octanoate	12.9 ^a^	191.53	263.78	282.73	323.31
34	Dimethyl trisulfide	0.36 ^a^	93.25	103.95	90.36	104.44
22	Ethyl hexanoate	55.3 ^a^	74.24	85.98	93.92	99.93
4	Ethyl acetate	32,600 ^a^	39.33	30.87	36.99	37.11
17	3-Methylbutyl acetate	93.9 ^a^	32.38	35.13	42.69	43.69
2	1,1-Diethoxyethane	2090 ^a^	24.44	31.31	28.84	30.67
9	Ethyl butanoate	81.5 ^a^	23.85	25.39	29.27	33.85
14	Hexanal	25.5 ^a^	20.46	16.72	29.67	23.92
6	Ethyl 2-methylpropanoate	57.5 ^a^	16.81	21.79	21.63	27.75
13	Ethyl 3-methylbutanoate	6.89 ^a^	15.62	18.53	20.03	33.46
30	Ethyl lactate	128,000 ^a^	13.37	11.30	12.42	10.67
18	Ethyl pentanoate	26.8 ^a^	10.95	10.71	19.15	12.99
43	1-Octen-3-ol	6.12 ^a^	10.27	7.74	12.97	16.57
11	Ethyl 2-methylbutanoate	18 ^a^	5.70	6.91	7.35	11.01
15	2-Methylpropanol	28,300 ^a^	3.72	4.16	4.64	4.17
56	Butanoic acid	964 ^a^	3.35	3.32	3.54	3.40
37	Acetic acid	160,000 ^a^	2.90	3.11	3.49	4.10
67	Pentanoic acid	389 ^a^	2.82	2.83	2.84	2.82
10	1-Propanol	54,000 ^a^	2.73	2.42	2.84	2.79
16	1-Butanol	2730 ^a^	2.50	2.74	3.01	3.21
77	Guaiacol	13.41 ^a^	2.46	5.33	2.25	3.86
59	Phenylacetaldehyde	262 ^a^	1.82	5.18	4.79	7.44
80	*β*-Ionone	1.3 ^b^	1.69	2.12	1.81	2.03
74	Hexanoic acid	2520 ^a^	1.54	1.55	1.60	1.68
63	3-Methylbutanoic acid	1050 ^a^	1.20	1.17	1.15	1.17
21	3-Methylbutanol	179,000 ^a^	1.18	1.32	1.46	1.38
51	2-Methylpropanoic acid	1580 ^a^	<1	1.07	1.11	1.06
31	1-Hexanol	5370 ^a^	<1	<1	1.14	1.20
7	2-Methylpropyl acetate	922 ^a^	<1	<1	<1	<1
35	Nonanal	122 ^a^	<1	<1	<1	1.14
86	Octanoic acid	2700 ^a^	<1	<1	<1	<1
1	2-Methylpropanal	1300 ^a^	<1	<1	<1	<1
87	4-Ethylphenol	123 ^a^	<1	<1	<1	<1
5	Ethyl propanoate	19,000 ^a^	<1	<1	<1	<1
49	Propanoic acid	18,200 ^a^	<1	<1	<1	<1
46	Decanal	70.8 ^a^	<1	<1	<1	<1
78	Ethyl 3-phenylpropanoate	125 ^a^	<1	<1	<1	<1
50	1-Octanol	1100 ^a^	<1	<1	<1	<1
58	1-Nonanol	806 ^a^	<1	<1	<1	<1
44	Furfural	44,000 ^a^	<1	<1	<1	<1
57	Ethyl decanoate	1120 ^a^	<1	<1	1.26	1.91
84	4-Ethylguaiacol	123 ^a^	<1	<1	<1	<1
79	2-Phenylethanol	28,900 ^a^	<1	<1	<1	<1
70	Ethyl phenylacetate	407 ^a^	<1	<1	<1	<1
62	2-Methylbutanoic acid	5932 ^a^	<1	<1	<1	<1
71	2-Phenylethyl acetate	909 ^a^	<1	<1	<1	<1
73	Ethyl dodecanoate	400 ^a^	<1	<1	<1	1.25
88	Decanoic acid	13,700 ^a^	<1	<1	<1	<1
81	4-Methylguaiacol	315 ^a^	<1	<1	<1	<1
27	2-Heptanol	1430 ^a^	<1	<1	<1	<1
64	Diethyl butanedioate	353,000 ^a^	<1	<1	<1	<1
48	Ethyl nonanoate	3150 ^a^	<1	<1	<1	<1
82	Phenol	18,900 ^a^	<1	<1	<1	<1
23	1-Pentanol	37,400 ^a^	<1	<1	<1	<1
32	2-Nonanone	483 ^a^	<1	<1	<1	<1
24	Hexyl acetate	5560 ^a^	<1	<1	<1	<1
25	3-Methylbutyl butanoate	915 ^a^	<1	<1	<1	<1
28	Ethyl heptanoate	13,200 ^a^	<1	<1	<1	<1
39	1-Heptanol	26,600 ^a^	<1	<1	<1	<1
60	Ethyl benzoate	1430 ^a^	<1	<1	<1	<1
68	Naphthalene	159 ^a^	<1	<1	<1	<1
40	3-Methylbutyl hexanoate	1400 ^a^	<1	<1	<1	<1
47	Benzaldehyde	4200 ^a^	<1	<1	<1	<1
85	Ethyl tetradecanoate	494,000 ^b^	<1	<1	<1	<1
19	2-Heptanone	Unknown ^c^	-	-	-	-
20	2-Pentylfuran	Unknown ^c^	-	-	-	-
41	Octyl acetate	Unknown ^c^	-	-	-	-
76	Geranylacetone	Unknown ^c^	-	-	-	-

^a^ Odor thresholds were taken from reference [40]. ^b^ Odor thresholds were taken from reference [41]. ^c^ Unknown, odor threshold was unavailable. -, not calculated.

## Data Availability

The data presented in this study are available on request from the corresponding author.

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
