# Peer review of "Comparison of the Aroma-Active Compounds and Sensory Characteristics of Different Grades of Light-Flavor Baijiu"

_foods, 2023, doi:10.3390/foods12061238_

Round 1

Reviewer 1 Report

·        The study compared the aroma compound content in four grades of Baijiu, however the increase in aroma content could be  have effect on consumers health, explain 

·        Lines 155-159: “Four trained panelists………..intensity value of the odor peak for each compound”.  In such a case, I believe E-nose detection is more reliable and accurate.

·        Lines 281-282“Several aroma-active ………. by the mass spectrum due to the low concentration.”, revise.

·        Lines 295- : “There were only a few compounds in the low-grade samples ……. values greater than 4”, explain why.

·        Some error typo should be adjusted (e,g  “β-Ionone” change to “β-Ionone”,…..etc).

Author Response

Response to Reviewer 1 Comments

Point 1: Lines 155-159: “Four trained panelists …… intensity value of the odor peak for each compound”. In such a case, I believe E-nose detection is more reliable and accurate.

Response 1: Thank you for your suggestion. The main purpose of this part was to identify aroma-active compounds. Nowadays, gas chromatography- olfactometry (GC–O) has been widely used to screen the aroma compounds and study the contribution of aroma compounds to the aroma profile of baijiu [1, 2].

Point 2: Lines 281-282 “Several aroma-active …… by the mass spectrum due to the low concentration.”, revise.

Response 2: Thank you for your comments. We have revised it into " A few aroma compounds could not be detected by mass spectrometry due to their low concentrations. But the panelists could clearly perceive the aroma." (Line 843-845)

Point 3: Lines 295- : “There were only a few compounds in the low-grade samples ……. values greater than 4”, explain why.

Response 3: Thank you for your suggestion. Subsequent research showed that the low content of aroma compounds in low-grade Baijiu might be the reason why the aroma intensity smelled by the panelists was low. In Results, we have speculated that it may be due to the different ageing years and blending proportion of their base liquors. (Line 935)

Point 4: Some error typo should be adjusted (e,g  “β-Ionone” change to “β-Ionone”,…..etc).

Response 4: Thank you for pointing it, we have made revisions in the manuscript.

[1]   WANG J, CHEN H, WU Y, et al. Uncover the flavor code of strong-aroma baijiu: Research progress on the revelation of aroma compounds in strong-aroma baijiu by means of modern separation technology and molecular sensory evaluation [J]. Journal of Food Composition and Analysis, 2022, 109

[2]   SONG H, LIU J. GC-O-MS technique and its applications in food flavor analysis [J]. Food Res Int, 2018, 114, 187-198.

Reviewer 2 Report

After carefully reading the manuscript entitled: " Comparison of the Aroma-active Compounds and Sensory Characteristic on Different Grades of the Light-flavor Baijiu" it can be concluded that the authors spent a lot of time and effort in conducting experiments and writing an article. The topic is interesting, but there are already published works with similar results. However, many things need to be worked through and explained. Below are remarks, clarification requests, and suggestions.

1. Extensive editing of the English language, grammatic, and style is required.

2. The sentences are too long, and very often, it is not easy to understand the meaning.

3. Kiloliters - it is not usual to express the amount of drink in this way. For strong alcoholic beverages, liters or hectoliters are most often used.

4. Rethink the name of the title. It does not fully reflect what was done in the paper.

5. References for the definition of the quality of Baijiu are missing.

6. Baijiu Samples. Detailed data of samples are missing. What were the base liquors used for blending? What was aging time, and etc. This data is essential for discussion, comparison, and conclusion.

7. Four trained panelists (25 years old on average) for GC–O-MS method. This method requires much experience.

8. Panel selection – is not according to ISO standards. Training with reference standards is not in line with current standards. (for example https://www.flavoractiv.com/spirit-sensory-solutions/; https://aroxa.com/spirits/; https://doemens.org/en/flavourstandards/ )

9. Some results are contradictory and not following literature data. Moreover, it does not align with the same authors' results presented in an article. For example, Concentrations of aroma compounds for Baijiu are 1000 times less in the present work than in Identification of Compounds Contributing to Trigeminal Pungency of Baijiu by Sensory Evaluation, Quantitative Measurements, Correlation Analysis, and Sensory Verification Testing in Journal of Agricultural and Food Chemistry.

Author Response

Response to Reviewer 2 Comments

Point 1: Extensive editing of the English language, grammatic, and style is required.

Response 1: Thank you for your suggestion. The manuscript has undergone English revisions and checked by a native English-speaker. If necessary, we can also find a language company to polish it.

Point 2: The sentences are too long, and very often, it is not easy to understand the meaning.

Response 2: Thank you for your comments. The manuscript has shortened some long sentences and revised the contents which are difficult to understand.

Point 3: Kiloliters - it is not usual to express the amount of drink in this way. For strong alcoholic beverages, liters or hectoliters are most often used.

Response 3: Thank you for your suggestion. We have changed "million kiloliters" into "billion liters" in the manuscript. (Line 34)

Point 4: Rethink the name of the title. It does not fully reflect what was done in the paper. 

Response 4: Thank you for your comments. This study comprehensively characterized and compared the aroma quality of different grades of LFB from the perspectives of aroma-active compounds and sensory characteristic. We had considered using "Comparison of the Aroma-active Compounds and Sensory Characteristic on Different Grades of the Light-flavor Baijiu by Gas Chromatography-Olfactometry-Mass Spectrometry, Quantitative Measurements and Sensory Evaluation" as the title. But considering that the title was too long, we chose the current title.

Point 5: References for the definition of the quality of Baijiu are missing.

Response 5: Thank you for your comments. We have cited relevant literature in the manuscript. (Line 36)

Point 6: Baijiu Samples. Detailed data of samples are missing. What were the base liquors used for blending? What was aging time, and etc. This data is essential for discussion, comparison, and conclusion.

Response 6: Thank you for your comments. We have added the aging time information of the samples’ base liquors to the manuscript (Line 292-294). Some discussions were made in the results section.

Point 7: Four trained panelists (25 years old on average) for GC–O-MS method. This method requires much experience.

Response 7: Thank you for your comments. The selected panelists had participated in professional GC-O training sessions and a lot of GC-O analysis before. They are competent for GC-O analysis.

Point 8: Panel selection – is not according to ISO standards. Training with reference standards is not in line with current standards. (for example https://www.flavoractiv.com/spirit-sensory-solutions/; https://aroxa.com/spirits/; https://doemens.org/en/flavourstandards/ )

Response 8: Thank you for your suggestion. Panel selection in this study was according to the guidelines of the ISO 8586:2012 standard. We have referred to this standard in the manuscript. (Line 707)

Point 9: Some results are contradictory and not following literature data. Moreover, it does not align with the same authors' results presented in an article. For example, Concentrations of aroma compounds for Baijiu are 1000 times less in the present work than in Identification of Compounds Contributing to Trigeminal Pungency of Baijiu by Sensory Evaluation, Quantitative Measurements, Correlation Analysis, and Sensory Verification Testing in Journal of Agricultural and Food Chemistry.

Response 9: Thank you for your suggestion. On the basis of the differences in manufacturing technique and aroma characteristics, Baijiu is generally divided into twelve typical aroma type. The content of aroma compounds in different flavor Baijiu has huge differences. The samples used in the literature were all strong-aroma type Baijiu, while the samples in this study were light-flavor Baijiu, so the contents were very different. The quantitative results of LFB in the literatures[1, 2] align with the results presented in this article. The results suggested the complexity and diversity of Baijiu flavor.

[1]   GAO W, FAN W, XU Y. Characterization of the key odorants in light aroma type chinese liquor by gas chromatography-olfactometry, quantitative measurements, aroma recombination, and omission studies [J]. J Agric Food Chem, 2014, 62(25), 5796-5804.

[2]   NIU Y, YAO Z, XIAO Q, et al. Characterization of the key aroma compounds in different light aroma type Chinese liquors by GC-olfactometry, GC-FPD, quantitative measurements, and aroma recombination [J]. Food Chem, 2017, 233, 204-215.

Reviewer 3 Report

This study is well planned, executed and written, following minor suggestions should be incorporated to further improve the contents of this study

In methodology section, ageing time of base material should be mentioned to improve the clarity of draft. At 4oC, how long liquor was stored. At what temperature samples were concentrated and how concentration was achieved under nitrogen? In MS section, please write split ratio and injection volume.

155-166, sensory related information is misplaced, it should be the part of sensory evaluation section

What gases were used to operate FID? Hydrogen and Oxygen? At what flow rate. Also write the name of software used the analysis of sensory data 

Author Response

Response to Reviewer 3 Comments

Point 1: In methodology section, ageing time of base material should be mentioned to improve the clarity of draft. At 4℃, how long liquor was stored. At what temperature samples were concentrated and how concentration was achieved under nitrogen? In MS section, please write split ratio and injection volume.

Response 1: Thank you for pointing it. Q1: We have added " The aging time of their base liquors was 0, 10, 20 and 30 years respectively." in line 272-273. Q2: The samples were stored at 4 ℃ from opening to the end of analysis, about 1 months. The concentration of aroma compounds in Baijiu samples under well sealed conditions has almost no significant change at 4 ℃. Q3: The extracts were concentrated at room temperature to the same volume (500 μL) under a gentle stream of nitrogen. Q4: We have explained in the manuscript that the injection volume was 1 μL, and the mode was splitless mode in line 590-591.

Point 2: 155-166, sensory related information is misplaced, it should be the part of sensory evaluation section

Response 2: Thank you for your comments. The purpose of this part is to identify aroma-active compounds, which could belong to the introduction of GC-O-MS method, so it was not included in the sensory evaluation part.

Point 3: What gases were used to operate FID? Hydrogen and Oxygen? At what flow rate. Also write the name of software used the analysis of sensory data

Response 3: Thank you for your suggestion. Q1: Hydrogen were used to operate FID at a flow rate of 30 mL/min. We have added it in line 632-633. Q2: We have mentioned and supplemented the name of software used in the analysis of sensory data in "2.6 Statistical Analysis" part (Line 812-814).

Round 2

Reviewer 2 Report

Dear authors, 

this revised version can be accepted.

Best regards, 

Author Response

Thanks for your recognition of our manuscript.